# Substantially High Hidden Blood Loss in Oblique Lateral Interbody Fusion: Retrospective Case Series

**DOI:** 10.3390/medicina58040527

**Published:** 2022-04-09

**Authors:** Koichiro Shima, Takashi Sono, Toshiyuki Kitaori, Kazutaka Takatsuka

**Affiliations:** 1Department of Orthopedic Surgery, Fukui Red Cross Hospital, Fukui 918-8501, Japan; seaman1991@gmail.com (K.S.); kitaorit0723@yahoo.co.jp (T.K.); mosashi@mx3.fctv.ne.jp (K.T.); 2Department of Orthopedic Surgery, Kitano Hospital, Osaka 530-8480, Japan

**Keywords:** hidden blood loss, oblique lateral interbody fusion, minimally invasive spine surgery

## Abstract

*Background and Objectives*: Measured blood loss frequently underestimates true blood loss; this discrepancy is called hidden blood loss (HBL). The purpose of the present study was to measure HBL in oblique lateral interbody fusion (OLIF). *Materials and Methods*: Patients who underwent two-stage OLIF at our institute from September 2017 to September 2021 were retrospectively reviewed. Total blood loss (TBL) and HBL were calculated using the gross formula. The age, sex, body mass index (BMI), operation time, measured blood loss, the number of fused segments, hematocrit (HCT), anticoagulant or platelet medication, blood transfusion, days of hospitalization, pre-/postoperative Japanese Orthopedic Association (JOA) score, and JOA recovery rate were compared. *Results*: A total of thirteen patients were included in the study. The average age, BMI, number of fused segments, operation time, estimated blood loss, and blood transfusion were 69.5 years, 23.3, 2.5, 250 min, 122 mL, and 230 mL, respectively. Five patients received anticoagulant or platelet therapy. Days of hospitalization, pre-/postoperative JOA score, and JOA recovery rate were 14.9 ± 5.1, 19.9 ± 2.7, and 18.0 ± 43.4%, respectively. The TBL and HBL were 688 and 797 mL, respectively. Stepwise multiple regression analysis revealed that younger age (*p* = 0.01), female sex (*p* = 0.01), and number of fused segments (*p* = 0.02) were significantly associated with higher HBL. *Conclusions*: The HBL in OLIF was 797 mL, which was more than other previously reported procedures. Therefore, OLIF may not be less invasive in terms of HBL. Blood loss after surgery should be considered, especially when patients are younger, are female, and have a greater number of fused segments.

## 1. Introduction

Measured blood loss frequently underestimates true blood loss in surgery, and this discrepancy is referred to as hidden blood loss (HBL). In hip fracture patients, HBL is associated with medical complications and increased hospital stay [1].

Lateral lumbar interbody fusions (LLIF) such as oblique lateral interbody fusion (OLIF) and extreme lateral interbody fusion (XLIF) are cutting-edge procedures for degenerative lumbar disease [2,3]. This procedure is minimally invasive because it uses short skin incisions, presents with less damage to paravertebral muscles, uses specific retractors, and causes less damage to bone. Our institution performed a two-staged operation: the OLIF procedure was performed in the first stage, and several days later, posterior pedicle screw fixation was performed, with or without decompression, in the second stage, after evaluating the effect of indirect neural decompression [3].

However, little is known about HBL in LLIF. Zhu et al. revealed that the average volume of HBL was 809 mL in single-level OLIF surgery, and the thickness of the abdominal wall’s soft tissue was a risk factor for HBL [4].

The purpose of the present study was to elucidate whether OLIF is minimally invasive in terms of HBL and to investigate other risk factors.

## 2. Materials and Methods

### 2.1. Patient Population

We retrospectively reviewed patients who underwent two-stage OLIF for degenerative lumbar disease at our institute from September 2017 to September 2021.

Patients who underwent single-stage OLIF and posterior spinal fixation were excluded.

We recorded age, sex, body mass index (BMI), operation time, measured blood loss, the number of fused segments, hematocrit (HCT), the use of anticoagulant or platelet medication, the amount of blood transfusion, the days of hospitalization, and the pre-/postoperative Japanese Orthopedic Association (JOA) score. JOA recovery rate was calculated as follows: (postoperative JOA score − preoperative JOA score)/(29 − preoperative JOA score) ×100.

### 2.2. OLIF Procedure

Patients were positioned in the true lateral decubitus position, and the target disc and vertebra were marked under fluoroscopy. A skin incision was made two fingers anterior from the anterior border of the disc. External oblique muscle, internal oblique muscle, transverse abdominal muscles, and transverse abdominal fascia were split, and the retroperitoneal space was exposed. Psoas major muscle was split at one-third and the disc space was exposed. The disc materials were removed, and the titanium-coated PEEK cage (Clydesdale, Medtronic Sofamor Danek Inc. 2600 Sofamore Danek Drive Memphis, TN 38132, US) was inserted. Suction drainage was not set when the bleeding during surgery was controlled.

Several days later, posterior fixation was performed according to the patient’s condition. Percutaneous pedicle screw (PPS) was used for four patients. Open pedicle screw fixation surgery, with or without L5/S interbody fusion, was performed on nine patients.

### 2.3. HBL Calculation

The intraoperative estimated blood loss (EBL) was estimated by the blood from the suction tube and the weight of the gauze.

The estimated blood volume was calculated based on sex, height (h), and weight (w), using the following previously reported formulas [5]:Female: EBV (L) = h^3^(m) × 0.356 + w(kg) × 0.033 + 0.183
Male: EBV (L) = h^3^ (m) × 0.367 + w(kg) × 0.032 + 0.604

Total blood loss (TBL) was calculated using preoperative and postoperative HCT and EBL. We denote HCTpre as preoperative HCT, and HCTpost as HCT three days after surgery. The formula was as follows:TBL (L) = EBV (L) × 2 × (HCTpre − HCTpost)/(HCTpre + HCTpost)

If patients received blood transfusion, then HCTpost was overestimated. Thus, we take into account the blood transfusion to the HBL. HBL was calculated as follows:HBL (mL) = TBL (mL) + Blood transfusion (mL) − EBL (mL)

### 2.4. Statistical Analysis

All the statistical analyses were performed with the JMP Pro ver. 13.0 software (SAS Institute Inc., Cary, NC, USA). Stepwise decreasing logistic regression was performed to explore factors associated with HBL. A *p*-value of <0.05 was determined to be statistically significant.

## 3. Results

A total of thirteen patients were included in the present study, comprising six males and seven females. The average age was 69.5, and the average BMI was 23.3. Five patients received anticoagulant or platelet therapy. During the operation, the average number of fused segments was 2.5, the average operation time was 250 min, the average EBL was 122 mL, and the average blood transfusion was 230 mL. As for blood test, the average preoperative HCT was 37.6, and the average postoperative HCT was 31.0. The calculated TBL was 688 ± 318 mL, and HBL was 797 ± 275 mL. For the clinical course, the duration of hospitalization was 47.3 ± 32.7 days, the pre-/postoperative JOA scores were 14.9 ± 5.1 and 19.9 ± 2.7, respectively, and the JOA recovery rate was 18.0 ± 43.4. Demographic data are shown in Table 1.

Stepwise multiple regression analysis demonstrated that that younger age (*p* = 0.01), female sex (*p* = 0.01), and a greater number of fused segments (*p* = 0.02) were significantly associated with higher HBL. (Table 2) There was no correlation between HBL and postoperative course.

## 4. Discussion

In the present study, the HBL in OLIF was 797 mL.

In a previously reported study, HBL measures in the anterior lumbar interbody fusion (ALIF), posterior spinal fusion (PSF), posterior lumbar fusion (PLF), burst fracture (BF), cervical laminoplasty (CLP), posterior lumbar interbody fusion (PLIF), and minimally invasive transforaminal lumbar interbody fusion (MIS-TLIF) procedures were 350, 600, 362, 303, 337, 419, and 488 mL, respectively [6,7,8,9,10,11,12]. These data are summarized in Table 3. Zhu et al. revealed that the average volume of HBL was 809 mL in single-level OLIF surgery [4], which is similar to the present study. Compared to other procedures, the HBL in the OLIF procedure was similar to or more than other procedures; although, the number of operated segments were different.

We considered several reasons for these results. First, there was some bleeding in the third space, such as retroperitoneal fat, after surgery. Second, although the conventional approach of the OLIF is between the psoas major muscle and the neurovascular band, i.e., the oblique lateral corridor, our approach was the trans-psoas approach, so there was some intramuscular bleeding. Third, we dissociated the opposite annulus fibrosus using a Cobb spinal elevator, which might have damaged the opposite psoas major muscle.

A previous study showed that the thickness of abdominal wall soft tissue was a risk factor of HBL [4]. The present study suggests that other risk factors—younger age, female sex, and a greater number of fused segments—were associated with higher HBL.

Miao et al. demonstrated that female patients had a risk of increased HBL in total hip arthroplasty (THA) [13]. The present study suggested the same result. The methods of calculating EBV differ between sexes, which may influence the difference between TBL and HBL between sexes.

Ju et al. illustrated that the length of surgery was a risk factor for increased HBL in ALIF [6], but Xu et al. demonstrated that multiple surgical levels were not associated with HBL in PLF [8]. We show that a greater number of fused segments were associated with a higher HBL. We believe that this is because surgeons usually open the retroperitoneal space and psoas major muscle according to a greater number of operated segments in OLIF procedures.

There was no correlation between HBL and postoperative course. This is because the invasiveness of the second stage of posterior surgery is different. It is necessary to analyze the HBL and postoperative course after matching the posterior procedure, such as PPS fixation.

There were several limitations in the present study. First, this was retrospective study, and the number of patients was very small. Further prospective studies that include a large number of patients are needed. Second, blood transfusion may influence HBL. We performed blood transfusion after the first operation in seven patients, and a previous study showed that HBL was positively associated with blood transfusion in THA [13].

In summary, the HBL in OLIF was 797 mL. OLIF may not be less invasive than other procedures in terms of HBL. We therefore recommend that surgeons take HBL into account after OLIF, even though measured blood loss is small, especially when patients are younger age, female, and/or undergo a greater number of fused segments.

## 5. Conclusions

The HBL in OLIF was 797 mL, which was more than what was reported in previously reported procedures. OLIF may not be less invasive in terms of HBL. We should therefore account for the blood loss after surgery, especially when patients are younger, are female, and have a greater number of fused segments.

## Figures and Tables

**Table 1 medicina-58-00527-t001:** Patient demographics.

Age (years)	69.5 ± 7.1
Sex (male/female)	6/7
BMI	23.3 ± 5.5
Operation Time (min)	250 ± 76
HCT pre (%)	37.6 ± 3.2
HCT post (%)	31.0 ± 3.5
EBL (mL)	122 ± 118
The number of fused segments	2.5 ±1.0
Anticoagulant medication (yes/no)	5/8
TBL (mL)	688 ± 318
HBL (mL)	797 ± 275
Transfusion (mL)	230 ± 243
Days of hospitalization (days)	47.3 ± 32.7
Preoperative JOA score	14.9 ± 5.1
Postoperative JOA score	19.9 ± 2.7
JOA recovery rate (%)	18.0 ± 43.4

BMI—body mass index; HCTpre—preoperative hematocrit; HCTpost—postoperative hematocrit; EBL—estimated blood loss; TBL—total blood loss; HBL—hidden blood loss; JOA—Japanese Orthopedic Association.

**Table 2 medicina-58-00527-t002:** Factors that influenced HBL in stepwise multiple regression analysis.

	*t* Value	*p* Value
Age	−3.64	0.01
Sex (female)	−3.73	0.01
Number of fused segments	3.13	0.02

**Table 3 medicina-58-00527-t003:** Comparison of OLIF with other procedures in terms of HBL.

Procedure	The Number of Operated Segments	Hidden Blood Loss (mL)	Author	Year
OLIF	2.5	797	The present study	2022
OLIF	1	809	Zhu	2021
PSF	NA	600	Smorgick	2013
ALIF	2.46	350	Ju	2016
PLF	1.6	362	Xu	2017
BF	1	303	Yin	2019
CLP	3	337	Jiang	2019
PLIF	1	419	Xu	2020
MIS-TLIF	1.39	488	Zhou	2020

OLIF—oblique lateral interbody fusion; ALIF—anterior lumbar interbody fusion; PSF—posterior spinal fusion; PLF—posterior lumbar fusion; BF—burst fracture; CLP—cervical laminoplasty; PLIF—posterior lumbar interbody fusion; MIS-TLIF—minimally invasive transforaminal lumbar interbody fusion; NA—not assessed.

## Data Availability

Not applicable.

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
