# Peer review of "Substantially High Hidden Blood Loss in Oblique Lateral Interbody Fusion: Retrospective Case Series"

_medicina, 2022, doi:10.3390/medicina58040527_

Round 1

Reviewer 1 Report

In the current study, authors have demonstrated the hidden blood loss in oblique lateral interbody fusion.

Here are my concerns about the study:

  1. There should be no period (.) in title.
  2. Authors have states that “To the best of our knowledge, however, HBL in LLIF is not yet investigated”. However, novelty of the study is compromised by below given recent studies:
  • Analysis of Hidden Blood Loss and its Risk Factors in Oblique Lateral Interbody Fusion Surgery. Clin Spine Surg. 2021 Nov 1;34(9):E501-E505, doi: 10.1097/BSD.0000000000001177.

This study on 70 patients have shown that substantial average HBL volume of 809 mL in single level OLIF surgery. The thickness of abdominal wall soft tissue was the risk factor of HBL.

Even in the minimally invasive surgery like MIS-TLIF, hidden blood loss has been reported, as in the following study:

  • Hidden blood loss and its possible risk factors in minimally invasive transforaminal lumbar interbody fusion. J Orthop Surg Res 15, 445 (2020). https://doi.org/10.1186/s13018-020-01971-5.

Likewise, a comprehensive search could be made and cited, and aim of paper should focus on adding to the consensus, as novelty is already compromised.

  1. All the manuscript is full of use of “we positioned”, “we made”, and “we split” in a single paragraph.
  2. Period (.) should always be after reference citations. Please check it throughout the manuscript.
  3. The use of English language in the manuscript is poor and should be rectified by native English speaker.

Author Response

Reviewer #1:

In the current study, authors have demonstrated the hidden blood loss in oblique lateral interbody fusion.

Here are my concerns about the study:

  • There should be no period (.) in title.

Thank you for your suggestion. I removed period in title. Additionally, other reviewer recommended changing the title, so I changed the title “Hidden Blood Loss of Oblique Lateral Interbody Fusion is substantially high”. (line 2-3)

  • authors have states that “To the best of our knowledge, however, HBL in LLIF is not yet investigated”. However, novelty of the study is compromised by below given recent studies:
  • Analysis of Hidden Blood Loss and its Risk Factors in Oblique Lateral Interbody Fusion Surgery. Clin Spine Surg. 2021 Nov 1;34(9):E501-E505, doi: 10.1097/BSD.0000000000001177.

This study on 70 patients have shown that substantial average HBL volume of 809 mL in single level OLIF surgery. The thickness of abdominal wall soft tissue was the risk factor of HBL.

Even in the minimally invasive surgery like MIS-TLIF, hidden blood loss has been reported, as in the following study:

  • Hidden blood loss and its possible risk factors in minimally invasive transforaminal lumbar interbody fusion. J Orthop Surg Res 15, 445 (2020). https://doi.org/10.1186/s13018-020-01971-5.

Likewise, a comprehensive search could be made and cited, and aim of paper should focus on adding to the consensus, as novelty is already compromised.

Thank you for your comment. I am sorry for not adequately searching the latest studies. I checked these papers and the paper referring to PLIF.

I changed introduction section at line 41-45 “However, little is known about HBL in LLIF. Zhu et al. revealed that the average volume of HBL was 809 mL in single-level OLIF surgery, and the thickness of the abdominal wall’s soft tissue was a risk factor for HBL [4]. The purpose of the present study is to elucidate whether OLIF is minimally invasive in terms of HBL and to investigate other risk factors.”

Additionally, I added the report of OLIF, MIS-TLIF and PLIF to Table 3.

Finally, I added discussion section in line 124-130 “In a previously reported study, HBL in the Anterior Lumbar Interbody Fusion (ALIF), Posterior Spinal Fusion (PSF), Posterior Lumbar Fusion (PLF), Burst Fracture (BF), Cervical Laminoplasty (CLP), Posterior Lumbar Interbody Fusion (PLIF), and Minimally Invasive Transforaminal Lumbar Interbody Fusion (MIS-TLIF) were 350, 600, 362, 303, 337, 419, and 488 ml, respectively [6-12]. These data are summarized in Table 3. Zhu et al. revealed that the average volume of HBL was 809 mL in single-level OLIF surgery [4], which is similar to the present study. and in line 139-141. A previous study showed that the thickness of abdominal wall soft tissue was a risk factor of HBL [4]. The present study suggests other risk factors: younger age, female sex, and a greater number of fused segments were associated with higher HBL..”

  • All the manuscript is full of use of “we positioned”, “we made”, and “we split” in a single paragraph.

I changed explanation in line 60-66 as follows.

“Patients were positioned in the true lateral decubitus position, and the target disc and vertebra were marked under fluoroscopy. A skin incision was made two fingers anterior from the anterior border of the disc. External oblique muscle, internal oblique muscle, transverse abdominal muscles, and transverse abdominal fascia were split, and the retroperitoneal space was exposed. Psoas major muscle was split at one-third and the disc space was exposed. The disc materials were removed, and the titanium-coated PEEK cage (Clydesdale, Medtronic Sofamor Danek Inc.) was inserted.”

  • Period (.) should always be after reference citations. Please check it throughout the manuscript.

Thank you for your suggestion. I changed the positions of periods as your suggestion.

  • The use of English language in the manuscript is poor and should be rectified by native English speaker.

Thank you for your comment. I sent the manuscript to the English editing service of MDPI.

Reviewer 2 Report

The main message of this article is the presence of unintended HBL in OLIF surgery compared to other surgical procedures. I would recommend changing the title to better describe this main point.

More importantly, had the relatively high HBL in OLIF affect postoperative course in the patients?  What was the clinical impact of HBL in OLIF surgery? More data and discussion needs to be addressed in this context.

Especially comparing low HBL versus High HBL might show differences in clinical improvement and days of hospitalization.   

Author Response

Reviewer #2:

The main message of this article is the presence of unintended HBL in OLIF surgery compared to other surgical procedures. I would recommend changing the title to better describe this main point.

Thank you for your recommendation. I changed the title to “Hidden Blood Loss of Oblique Lateral Interbody Fusion is substantially high” in line 2-3.

More importantly, had the relatively high HBL in OLIF affect postoperative course in the patients?  What was the clinical impact of HBL in OLIF surgery? More data and discussion needs to be addressed in this context.

Especially comparing low HBL versus High HBL might show differences in clinical improvement and days of hospitalization. 

Thank you for your comment. I checked days of hospitalization, preoperative / postoperative JOA score and JOA recovery rate. There was no correlation between HBL and postoperative course.

I added abstract section in line 14-15 “days of hospitalization, pre-/post-operative Japanese Orthopedic Association (JOA) score, and JOA recovery rate were compared.” and in line 19-20 “Days of hospitalization, pre-/post-operative JOA score, and JOA recovery rate were 14.9 ± 5.1, 19.9 ± 2.7, and 18.0 ± 43.4 %, respectively.”, material and method section in line 54-57 “the days of hospitalization, and the pre-/post-operative Japanese Orthopedic Association (JOA) score. JOA recovery rate was calculated as follows: (postoperative JOA score - preoperative JOA score) / (29 - preoperative JOA score) * 100.” and result section in line 104-106 “For the clinical course, the days of hospitalization was 47.3 ± 32.7 days, the pre-/post-operative JOA scores were 14.9 ± 5.1 and 19.9 ± 2.7, respectively, and the JOA recovery rate was 18.0 ± 43.4.” and “There was no correlation between HBL and postoperative course” in line 114-115. I also added these data to Table 1.

Considering these data, I changed Discussion section in line 152-155 “There was no correlation between HBL and postoperative course. This is because the invasiveness of the second stage of posterior surgery is different. It is necessary to analyze the HBL and postoperative course after matching the posterior procedure such as PPS fixation.

Round 2

Reviewer 2 Report

Although authors have revised their title according to the previous comment, the new title has potential redundancy in English. I would propose to change the title to  "A Substantially High Hidden Blood Loss in Oblique Lateral Interbody Fusion: Retrospective Case Series".

Author Response

# Reviewer 2

Although authors have revised their title according to the previous comment, the new title has potential redundancy in English. I would propose to change the title to  "A Substantially High Hidden Blood Loss in Oblique Lateral Interbody Fusion: Retrospective Case Series".

Thank you for your recommendation. I changed the title of this manuscript again.